# CBFB cooperates with p53 to maintain *TAp73* expression and suppress breast cancer

**Navdeep Malik**[1], **Hualong Yan**[1], **Howard H. Yang**[2], **Gamze Ayaz**[1], **Wendy DuBois**[1], **Yu-Chou Tseng**[1], **Young-Im Kim**[1], **Shunlin Jiang**[1], **Chengyu Liu**[3], **Maxwell Lee**[2], **Jing Huang**[1] *

**1** Cancer and Stem Cell Epigenetics Section, Laboratory of Cancer Biology and Genetics, Center for Cancer Research, National Cancer Institute, Bethesda, Maryland, United States of America, **2** High-Dimension Data Analysis Group, Laboratory of Cancer Biology and Genetics, Center for Cancer Research, National Cancer Institute, Bethesda, Maryland, United States of America, **3** Transgenic Core, National Heart, Lung, and Blood Institute, Bethesda, Maryland, United States of America

* huangj3@mail.nih.gov

**Data Availability Statement:** RNA-seq and ChIP-seq datasets of MCF10A WT and p53_KO cells have been deposited into the GEO with an accession number of GSE146278 and GSE169716, respectively. The links are as follows: https://www.ncbi.nlm.nih.gov/geo/query/acc.cgi?acc=

## Abstract

The *CBFB* gene is frequently mutated in several types of solid tumors. Emerging evidence suggests that CBFB is a tumor suppressor in breast cancer. However, our understanding of the tumor suppressive function of CBFB remains incomplete. Here, we analyze genetic interactions between mutations of *CBFB* and other highly mutated genes in human breast cancer datasets and find that *CBFB* and *TP53* mutations are mutually exclusive, suggesting a functional association between CBFB and p53. Integrated genomic studies reveal that *TAp73* is a common transcriptional target of CBFB and p53. CBFB cooperates with p53 to maintain *TAp73* expression, as either CBFB or p53 loss leads to TAp73 depletion. TAp73 re-expression abrogates the tumorigenic effect of CBFB deletion. Although TAp73 loss alone is insufficient for tumorigenesis, it enhances the tumorigenic effect of NOTCH3 over-expression, a downstream event of CBFB loss. Immunohistochemistry shows that p73 loss is coupled with higher proliferation in xenografts. Moreover, TAp73 loss-of-expression is a frequent event in human breast cancer tumors and cell lines. Together, our results signifi-cantly advance our understanding of the tumor suppressive functions of CBFB and reveal a mechanism underlying the communication between the two tumor suppressors CBFB and p53.

## Author summary

The success of precision medicine in oncology requires a detailed understanding of genetic alterations and the functional associations between them. Recent genome-wide sequencing studies found that breast, ovarian, and prostate tumors have frequent *CBFB* mutations. Emerging evidence suggests that CBFB is a tumor suppressor in breast tumors. However, our understanding of the tumor suppressive functions of CBFB remains frag-mented. In this study, our genetic analyses of *CBFB* and *TP53* mutations suggest that CBFB and p53 are functionally associated in breast tumors. By leveraging the rich

GSE146278 and https://www.ncbi.nlm.nih.gov/geo/query/acc.cgi?acc=GSE169716. RNA-seq data of MCF10A WT, CBFB_KO, RUNX1_KO cells are public in the GEO (GSE120216). RUNX1 ChIP-seq is public in the GEO (GSE129314). The underlying numerical data for all of the graphs and summary statistics are either in main figures or Supporting Information.

**Funding:** This work was supported by intramural grants, 1ZIABC011158-12 and 1ZIABC011504-08, at the National Cancer Institute (NCI), USA, to JH. The funders have no role in the study design, data collection and analysis, decision to publish, or preparation of the manuscript.

**Competing interests:** The authors have declared that no competing interests exist.

knowledge of the tumor suppressive function of p53, we found that CBFB induces the expression of *TAp73*, a well-established mediator for the tumor suppressive function of p53. TAp73 re-expression inhibits the tumorigenicity of CBFB-deficient breast cells. In addition, TAp73 depletion drives breast tumorigenesis by cooperating with NOTCH3, a CBFB-repressed oncogene. Moreover, human breast tumors and cancer cell lines frequently lose the expression of TAp73. Together, our study gains a mechanistic understanding of the tumor suppressive functions of CBFB, reveals a functional association between CBFB and p53 in breast cancer, and has important implications in precision medicine for breast cancer.

## Introduction

Breast cancer is the second leading cause of cancer-related death among women. It is estimated that about 270,000 new cases and 42,000 deaths from breast cancer occurred in the United States in 2019 [1]. Although the breast cancer death rate has been steadily declining for the past three decades, the decline rate has slowed in recent years [2], suggesting that new treatments are needed to reduce breast cancer mortality further. Precision medicine, which applies targeted therapeutics based on the identification of patient subgroups using molecular profiling, holds great potential in oncology, including breast cancer. However, the ultimate clinical application of precision medicine requires a deeper mechanistic understanding of the functional consequences of each mutation [3,4].

Recent genome-wide sequencing studies have discovered many genetic alterations in breast cancer genomes [5,6]. Although most alterations occur in well-studied cancer genes, such as *TP53*, *GATA3*, and *PIK3CA*, a significant portion of them occur in previously under-appreciated genes, such as *CBFB* (Core-Binding Factor Subunit Beta), with mutations found in about five percent of human breast tumors. Understanding whether and how these under-studied genes contribute to breast cancer pathogenesis is critical for developing novel strategies to treat breast cancer with greater precision. Despite the recent developments in computational methods, it remains challenging to distinguish driver cancer genes from passengers and determine their functions in breast cancer etiology [7,8].

CBFB is historically regarded as a transcriptional co-factor for RUNX family member proteins, RUNX1, RUNX2, and RUNX3. The well-accepted model for CBFB function states that CBFB enhances the chromatin binding of the RUNX proteins by heterodimerizing with them to form a transcriptional complex, regulating the expression of genes with diverse functions in many cells and tissues [9]. In acute myeloid leukemia (AML), *CBFB* is one of the most targeted genes, with most of the genetic alterations being fusions, such as to the *MYH11* gene [10]. However, in breast cancer, point mutations are the dominant type of alteration. These observations suggest that CBFB may have differential roles in AML and breast cancer. Indeed, we have recently discovered a novel function for CBFB in translation regulation in breast cancer [11]. In breast cells, CBFB regulates both translation and transcription to suppress tumorigenesis [11,12]. CBFB enhances the translation of hundreds of mRNAs, including the *RUNX1* mRNA. In the nucleus, the CBFB-RUNX1 transcriptional complex represses *NOTCH3*, an oncogene in breast cancer. However, *NOTCH3* overexpression does not fully recapitulate the effect of *CBFB* loss-of-function in breast cancer [11], suggesting that other mechanism(s) also contribute to the tumor suppressive function of CBFB. It is also unknown whether the CBFB/RUNX1 axis cooperates with different pathways to suppress breast cancer.

As the guardian of the genome, p53 has been well-studied in many types of cancer, including breast cancer. Mutations of the *TP53* gene occur in about one-third of human breast tumors. After p53 is activated by various stresses, such as DNA damage insults or oncogene activation, it binds to chromatin and regulates the transcription of numerous targets that elicit different cellular outcomes, such as cell cycle arrest, apoptosis, and DNA repair [13]. The relative contribution of individual target genes and cellular pathways in mediating the tumor suppressive function of p53 remains an active research area and attracts significant attention.

p53 has two homologs, p63 and p73 [14,15]. Both are critical mediators for p53-induced apoptosis upon DNA damage [16–19]. Interestingly, p73 is a transcriptional target of p53 [20], suggesting an interconnected regulatory network formed by the p53 family members. The *TP73* gene encodes several isoforms with the TAp73 isoforms and the ΔNp73 isoforms being driven by two different promoters [15]. The TAp73 isoforms, but not the ΔNp73 isoforms, contain an amino-terminal transactivation (TA) domain [15]. Studies show that each isoform has context-dependent functions in different cells and tissues [14]. Overexpression of TAp73 induces apoptosis in cells [21]. In developing neurons, ΔNp73 is the predominant p73 isoform and performs an anti-apoptotic function, presumably by interfering with p53 and/or TAp73 [22]. Generally, ΔNp73 has oncogenic roles, while TAp73 has anti-transformation roles [23,24]. In human mammary epithelial cells, TAp73, but not ΔNp73, blocks epithelial-to-mesenchymal transition [25]. Isoform-specific knockout mice show that TAp63 and TAp73 isoforms have tumor-suppressive functions. ΔNp63 and ΔNp73 knockout mice have developmental defects in the nervous system, limbs, and skin [14,26–28]. It is worth noting that recent studies revealed that TAp73 can maintain tumor progression in medulloblastoma, lung cancer, osteosarcoma, and breast cancer stem-like cells, suggesting that the roles of TAp73 in cancer are more complex than originally thought [29–31].

The goal of this study is to further investigate the tumor suppressive function of CBFB. We applied an algorithm that calculates the genetic interactions of *CBFB* and other cancer genes in breast cancer. We found that *CBFB* and *TP53* mutations are mutually exclusive, suggesting that CBFB communicates with p53 in breast cancer cells. Built upon this genetic discovery, we used genomics tools to identify shared downstream targets of CBFB and p53. We focused on one of the targets, *TP73*. Specifically, CBFB and p53 cooperatively induce *TAp73*, but not *ΔNp73*. TAp73 loss alone is not sufficient to generate breast tumors; however, it enhances the pro-tumorigenic ability of NOTCH3. Furthermore, we found that TAp73 loss is a frequent event in both human breast tumors and cell lines. Our study significantly advances our understanding of the tumor suppressive function of CBFB and reveals a functional association between CBFB and p53 in breast cancer cells.

## Results

### *CBFB* and *TP53* mutations are mutually exclusive

To further explore the tumor suppressive function of CBFB in breast cancer, we calculated the genetic interaction (epistasis) of the mutations of two cancer genes in breast cancer. The general assumption for this analysis is that if the mutations of two genes genetically interact (meaning mutually exclusive or co-occurring), the proteins encoded by these two genes likely functionally communicate [32,33]. Moreover, a recent study demonstrated that tumor subtypes and tumor mutational load (TML) could also cause mutual exclusivity of two gene mutations [34]. Based on these assumptions, we calculated the genetic interaction between mutations in the *CBFB* gene and other highly mutated genes in the METABRIC (Molecular Taxonomy of Breast Cancer International Consortium) dataset [35]. We identified nine genes that genetically interact with *CBFB* (Fisher's exact test, p-value <0.1) (Fig 1A, upper panel).

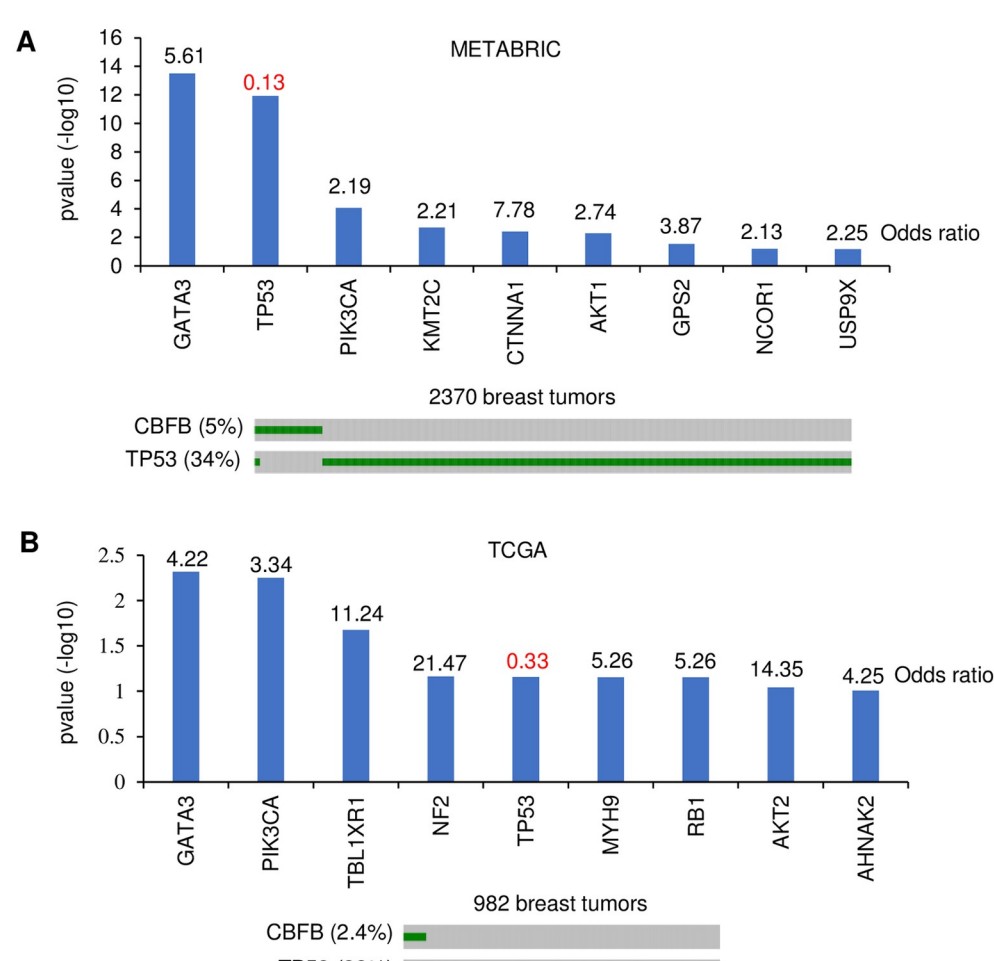

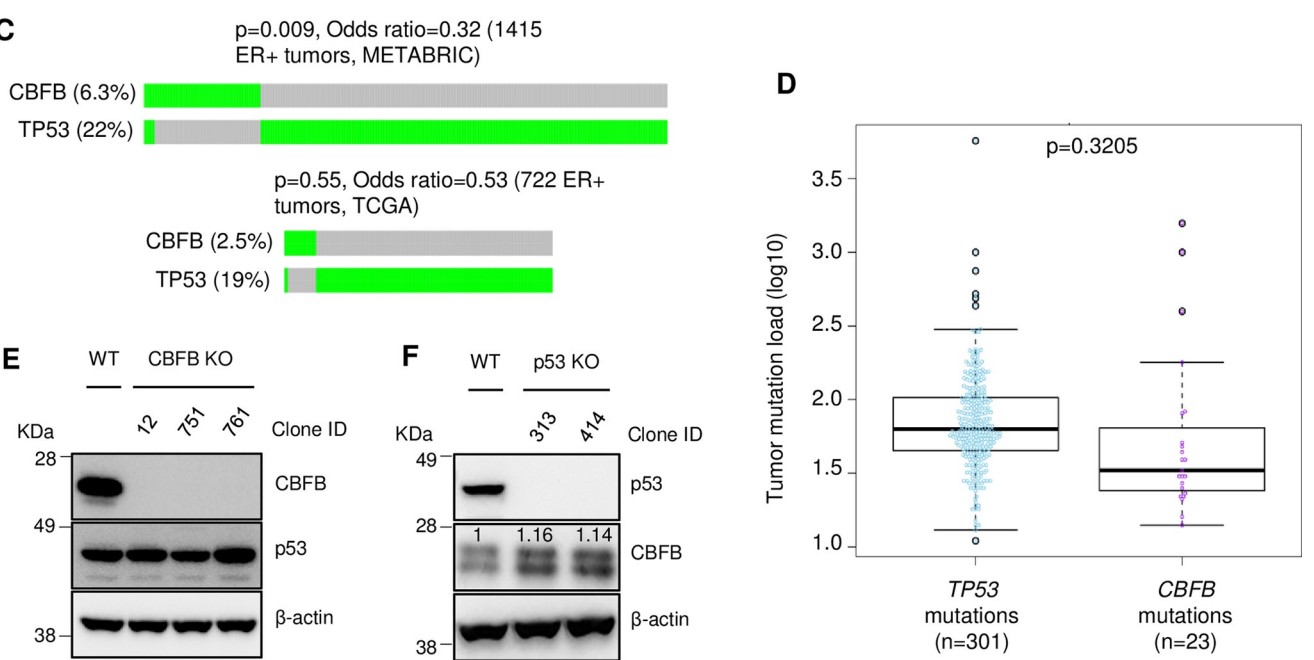

**Fig 1. Mutations in *CBFB* and *TP53* are mutually exclusive.** (**A-B**) Analyses of the genetic interaction between *CBFB* mutations and other gene mutations in the METABRIC (**A**) and TCGA (**B**) datasets. Upper panels: Odds ratios (numbers above bars) and p-values are from Fisher's exact test. Only genetic interactions with a p-value of less than 0.1 are shown. An odds ratio larger than 1 indicates the co-occurrence of two gene mutations. In contrast, an odds ratio of less than 1 (in red) means that two gene mutations are mutually exclusive. Lower panels: Oncoprints of *CBFB* and *TP53* mutations in the METABRIC and TCGA datasets. The total numbers of tumors in the dataset are shown. Only tumors with mutations in at least one gene are shown in the oncoprints. (**C**) Oncoprints show *CBFB* and *TP53* mutations in ER+ tumors in the METABRIC and TCGA datasets. p-values and odds ratios are from Fisher's exact test. (**D**) Tumor mutation load of breast tumors carrying *TP53* or *CBFB* mutations. The p-value is from the Wilcoxon test. (**E**) Immunoblotting (IB) shows the expression of CBFB, p53, and β-actin in the wild type (WT) and *CBFB* knockout (KO) MCF10A cells. (**F**) IB shows the expression of CBFB, p53, and β-actin in the WT and *TP53* KO MCF10A cells. The numbers in the image of CBFB blot are CBFB intensity normalized to β-actin intensity.

Among these nine genes, only *TP53's* mutations were mutually exclusive with *CBFB* mutations (odds ratio <1), while other genes' mutations were co-occurring (odds ratio >1). Using an independent breast cancer dataset in TCGA (The Cancer Genome Atlas), we identified nine genes that genetically interact with *CBFB* (Fig 1B, upper panel). The gene list partially overlaps with that generated from METABRIC, suggesting that the tumor selection criteria of these two datasets affect the results of this analysis. Using the cBioPortal [36], we confirmed that *CBFB* and *TP53* mutations were mutually exclusive in both the METABRIC and TCGA datasets (Fig 1A and 1B, lower panels). Thus, we decided to focus on the mechanisms underlying the mutual exclusivity of *CBFB* and *TP53* mutations.

*TP53* mutations are enriched in estrogen receptor (ER)-negative breast tumors, while *CBFB* mutations are generally associated with ER-positive subtype [37]. Therefore, one possibility is that CBFB and *TP53* mutations' mutual exclusivity is caused by their differential enrichment in ER-positive and ER-negative tumors. To test this possibility, we performed epistasis analysis in ER-positive breast tumors in the METABRIC dataset and still observed mutual exclusivity (p = 0.0009, odds.ratio = 0.32, CI = 0.134, 0.679) of *CBFB* and *TP53* mutations (Fig 1C). In the TCGA dataset, although *CBFB* and *TP53* mutations in ER-positive tumors trended toward mutually exclusive (odds.ratio = 0.53), the observation was not statistically significant (p = 0.55), which may be due to the relatively small number of tumors in the dataset (Fig 1C). These results suggest that even if tumor subtype is involved, other mechanisms contribute to CBFB and TP53 mutationsmutual exclusivity. A second possibility is that TML underlies the mutual exclusivity of *CBFB* and *TP53* mutations. To test this, we calculated TML in tumors carrying either *CBFB* mutations or *TP53* mutations. We found no statistically significant difference in TML between these two groups (Fig 1D). Therefore, TML is unlikely to cause the mutual exclusivity of mutations of these two genes. We then explored the possibility that CBFB and p53 crosstalk. Given that p53 has been well-studied as a tumor suppressor in cancer, understanding the mechanism(s) underlying this putative functional association between *CBFB* and *TP53* mutations will illuminate the tumor suppressive function of CBFB in breast cancer.

## CBFB and p53 share a set of transcriptional targets

One scenario of the functional association between CBFB and p53 is that these two proteins maintain each other's expression. To test this, we generated *CBFB* and *TP53* knockout (CBFB_KO and p53_KO) MCF10A cells using CRISPR (clustered regularly-interspaced short palindromic repeats) technology (Fig 1E and 1F). Immunoblotting results showed that the loss of CBFB did not alter the level of p53 (Fig 1E), while CBFB levels slightly increased in p53_KO cells (Fig 1F). Thus, the deletion of CBFB and p53 does not decrease the expression of the other.

A second scenario is that CBFB and p53 cooperatively regulate the same gene or pathway. In this context, mutation of either *CBFB* or *TP53* leads to the dysregulation of the gene or

pathway. Previously, we showed that CBFB is a dual-function protein that regulates both translation and transcription in breast cancer [11]. The tumor suppressive functions of p53 are primarily attributed to its transcriptional activity [13]. Therefore, we hypothesized that CBFB and p53 regulate a single or a set of transcriptional targets that mediate their tumor suppressive functions. To test this hypothesis, we used RNA-seq to identify common transcriptional targets of CBFB and p53 in MCF10A cells. Transcriptional targets of CBFB were previously identified [11]. When CBFB functions as a transcriptional co-factor, it heterodimerizes with RUNX1 to form the CBFB/RUNX1 transcription complex. Therefore, to identify direct transcriptional targets of CBFB, we concentrated on the common targets of CBFB and RUNX1. We compared gene expression profiles of parental MCF10A cells with CBFB KO or RUNX1 KO cells (Fig 2A). This analysis revealed 212 common targets regulated by both CBFB and RUNX1. Using RNA-seq, we identified 779 p53 targets by comparing the gene expression profiles of parental MCF10A cells and p53 KO cells. The CBFB/RUNX1 complex and p53 shared 48 transcriptional targets (Fig 2A). Because the basis of our hypothesis is that both the CBFB/RUNX1 complex and p53 are tumor suppressors for breast cancer, we focused on 33 shared targets that were regulated by the CBFB/RUNX1 complex and p53 in the same direction. Among these 33 targets, 17 genes were up-regulated and 16 down-regulated by both the CBFB/RUNX1 complex and p53 (Fig 2B). Pathway analyses using DAVID Bioinformatics [38] and Ingenuity Pathway Analysis did not identify any biologically relevant pathways among these 33 targets. Therefore, we examined these genes individually.

The premise of our study was to exploit the well-studied p53 gene network to gain insights into the tumor suppressive function of CBFB. Thus, we were particularly interested in *TP73*, a well-established mediator for the p53-regulated DNA damage response and, in some contexts, the tumor suppressive function of p53 [16,20,26]. ChIP-seq analysis revealed the binding of RUNX1 and p53 in an intragenic region of the *TP73* locus (Fig 2C), suggesting that *TP73* is a direct transcriptional target of CBFB/RUNX1 and p53. To confirm this possibility, we amplified a 501-base pair (bp) DNA fragment surrounding the binding site as a putative RUNX1 full-length response element (FL_RE) and cloned this fragment into a luciferase reporter (Fig 2C). The luciferase assay showed that RUNX1 induced the luciferase activity through FL_RE, suggesting that *TP73* is a direct transcriptional target of the CBFB/RUNX1 complex (Fig 2D). The luciferase activity of FL_RE was higher in CBFB WT cells than in KO cells (Fig 2E), consistent with the model that CBFB facilitates RUNX1's transcriptional activity. To narrow down the response region, we generated two truncated fragments from FL_RE: one is 175 bp (fragment1), and another is 431 bp (fragment2) (Fig 2C). The luciferase assay showed that fragment2 but not fragment1 was responsive to RUNX1, indicating that the response region falls within the fragment from 175 to 431 bp (Fig 2F). RUNX1 binding motif is CCnC, in which C is cytosine and n is A, T, or G [39]. After scanning the 175–431 region, we identified three putative RUNX1 motifs (Fig 2D). We mutated either a single or combination of these motifs in the reporter and performed the luciferase assay (Fig 2G). Our result showed that mutating motif 2 or 3 alone decreased the luciferase activity. However, the contribution of motif 1 became prominent only in combination with motif 2, 3 or both (compare the luciferase activity of Mut1 to control, Mut1_2 to Mut2, Mut1_3 to Mut3, or Mut1_2_3 to Mut2_3). Therefore, all three motifs mediated the response to RUNX1, and the second motif was the primary one in the context of single-motif mutation. Mutation of all three motifs completely abolished the response to RUNX1 (Fig 2G). We then examined the cooperation of RUNX1 and p53 in the induction of TAp73. The luciferase activity of RUNX1 and p53 co-transfection was significantly higher than that of single gene transfection (Fig 2H), suggesting that RUNX1 and p53 cooperate in p73 induction. Together, our results establish that *TP73* is a *bona fide* transcription target of the CBFB/RUNX1 complex.

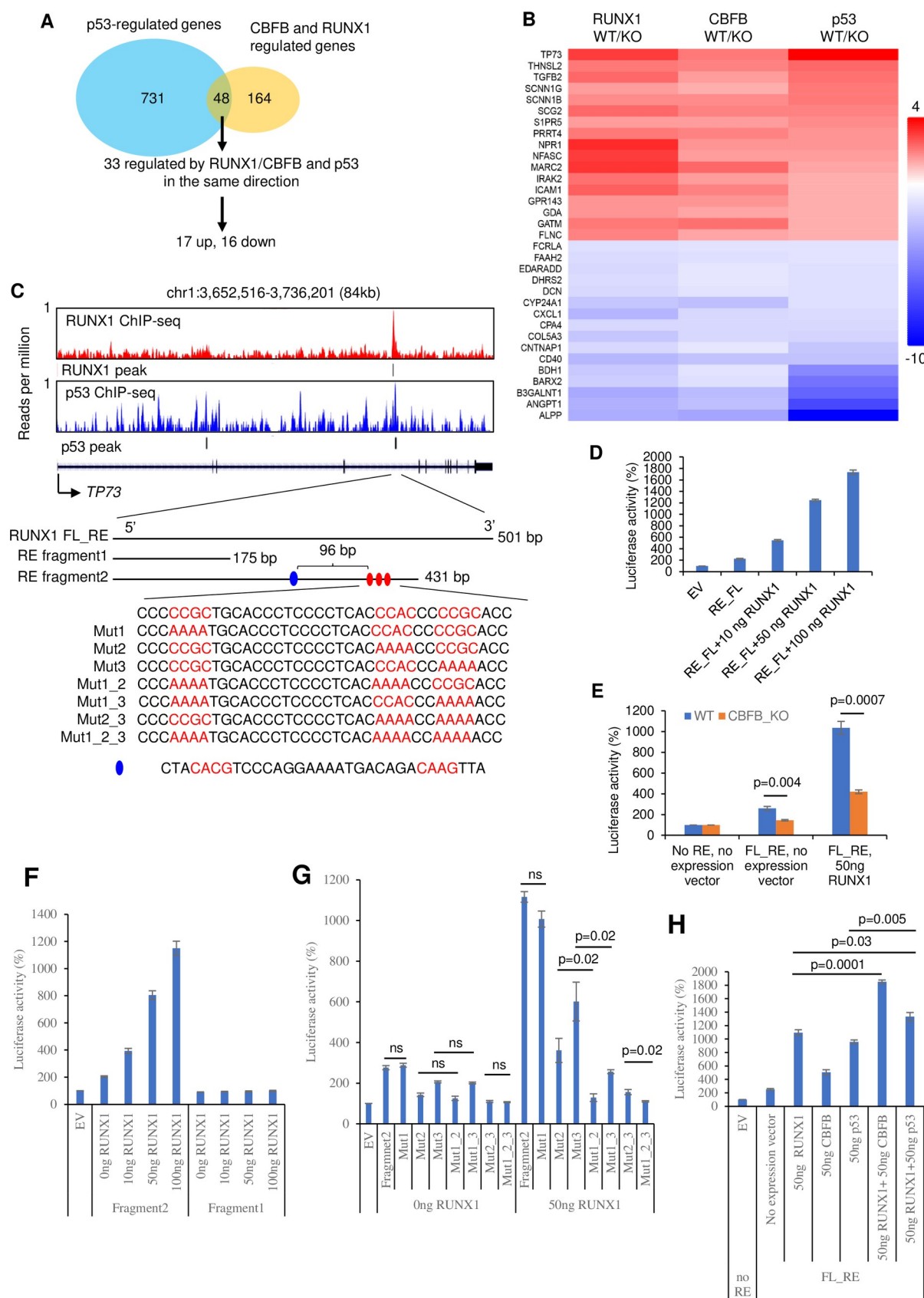

**Fig 2. *TP73* is a common transcriptional target of the CBFB/RUNX1 complex and p53.** (**A**) A flow chart shows the strategy to identify common transcriptional targets of the CBFB/RUNX1 complex and p53 in MCF10A cells using RNA-seq. (**B**) A heatmap showing the identified 33 common targets of the CBFB/RUNX1 complex and p53. (**C**) Genomic views of ChIP-seq showing the binding of RUNX1 and p53 to the *TP73* gene locus. Putative RUNX1 response elements (RE): FL_RE, full-length RE (501 bp); RE fragment1, a fragment of 175 bp; RE fragment2, a fragment of 431 bp. The three red ovals indicate the putative RUNX1 motifs, and the blue oval the putative p53 consensus motif. Sequences of different mutated versions of REs are shown with putative motifs highlighted in red. (**D**) Luciferase assays in MCF10A cells. EV, empty reporter vector (EV); FL_RE, a reporter containing full-length RE. A plasmid expressing RUNX1 at various concentrations is co-transfected with the reporter. (**E**) Luciferase assays in WT and CBFB_KO MCF10A cells. (**F**) Luciferase assays using EV, RE fragment1, or RE fragment2 co-transfected with a RUNX1-expressing plasmid at a concentration indicated in MCF10A cells. (**G**) Luciferase assays using EV, RE fragment2, or fragment2 containing various mutations (designated in **C**) co-transfected with 0 ng or 50 ng of a RUNX1-expressing plasmid in MCF10A cells. (**H**) Luciferase assays examine the cooperativity of RUNX1, p53, and CBFB. For **D**-**H**, luciferase activities are the normalized values (see Materials and Methods). The value of EV is set to 100%. Error bars = SEM; n = 3 biological repeats; p-values are t-test (two-tailed, two-sample with equal variance).

## TAp73, not ΔNp73, is regulated by CBFB

The *TP73* gene encodes several p73 isoforms, including *TAp73* and *ΔNp73*, driven by two different promoters [15]. Thus, the 5' end is the critical region for distinguishing *TAp73* from *ΔNp73*. However, our RNA-seq is biased to the 3' end of RNAs and cannot distinguish these two isoforms. To overcome this, we utilized an epigenetic mark, histone H3 lysine 4 trimethylation (H3K4me3), to demarcate the promoters for *TAp73* and *ΔNp73* (Fig 3A). We detected the H3K4me3 signal at both promoters using ChIP-seq (Fig 3A), indicating that both isoforms are expressed. To explore whether CBFB, RUNX1, and p53 regulate *TAp73 and ΔNp7*3, we also performed H3K4me3 ChIP-seq in CBFB_KO, RUNX1_KO, and p53_KO cells. The deletion of CBFB, RUNX1, or p53 resulted in a decreased H3K4me3 signal at the *TAp73* promoter. In p53_KO cells, there was a slight reduction of the H3K4me3 signal at the *ΔNp73* promoter (Fig 3A). Using TAp73- and ΔNp73-specific antibodies, we found that the deletion of *CBFB*, *RUNX1*, and *TP53* reduced the levels of TAp73 (Fig 3B and 3C) but not ΔNp73 (Fig 3D–3F). Therefore, TAp73 is the main target of CBFB, RUNX1, and p53.

Notably, the reduction in TAp73 expression upon *CBFB* or *RUNX1* deletion is reversible, as *CBFB* overexpression in CBFB_KO cells or *RUNX1* overexpression in RUNX1_KO cells restored the levels of TAp73 (Fig 3G and 3H). Therefore, the loss of TAp73 is not caused by a secondary genetic alteration following *CBFB* or *RUNX1* deletion.

## TAp73 loss facilitates but is insufficient for the transformation of breast cells

CBFB or RUNX1 deletion transforms MCF10A cells [11]. To determine whether TAp73 loss is involved in the transformation, we overexpressed *TAp73* in CBFB_KO and RUNX1_KO MCF10A cells using a lentiviral vector (Fig 4A) and conducted the anchorage-independent growth assay. TAp73 overexpression decreased the transformation ability of CBFB_KO and RUNX1_KO MCF10A cells, judged by the percentage of transformed cells and the size of the colonies (Figs 4B, 4C and S1), suggesting that TAp73 loss facilitates the transformation of CBFB_KO and RUNX1_KO MCF10A cells. We then tested whether p73 loss alone is sufficient for the transformation of the cells. To this end, we generated p73_KO MCF10A cells using CRISPR-Cas9 (Fig 4D). p73 deletion did not alter the levels of p53, CBFB, and RUNX1, ruling out a feedback loop under this condition. In the anchorage-independent assay, p73_KO clones did not form colonies, suggesting that TAp73 deletion alone is insufficient for the transformation (Fig 4E and 4F). This phenotype is similar to that of p53 deletion, which is also not enough to transform cells [40].

CBFB_KO and RUNX1_KO MCF10A cells form tumors in immunocompromised NSG (NOD-scid-gamma) mice [11]. To investigate the effect of TAp73 on the tumorigenicity of

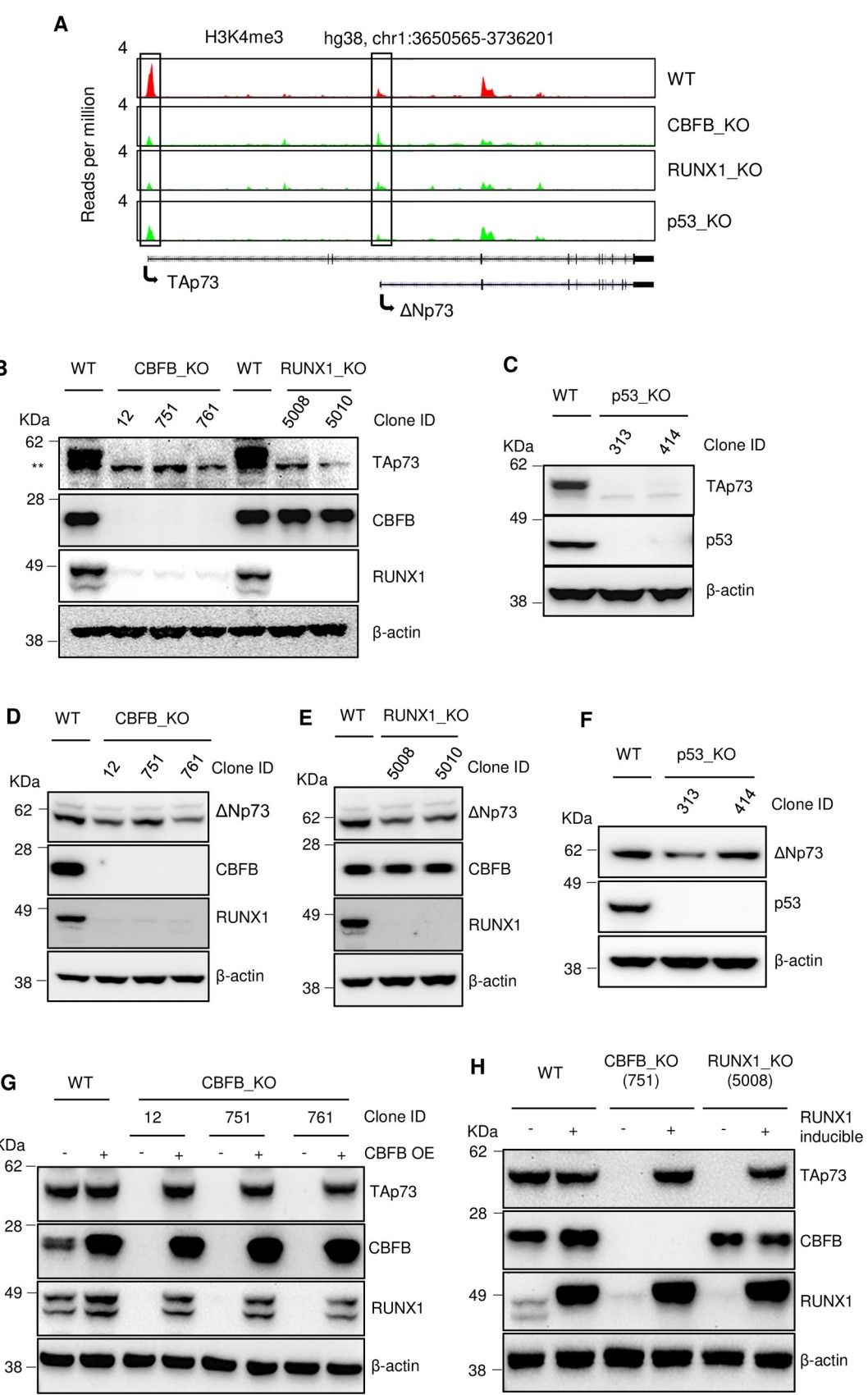

**Fig 3. The CBFB/RUNX1 complex and p53 induce TAp73 but not ΔNp73.** (**A**) ChIP-seq showing the effects of CBFB, RUNX1, or p53 deletion on H3K4me3 levels (a marker for promoters) near the promoters (marked by two boxes) for *TAp73* and *ΔNp73* in MCF10A cells. (**B**) IB shows the effects of *CBFB* or *RUNX1* deletion on TAp73 protein levels in MCF10A cells. (**C**) IB shows the impact of *TP53* deletion on TAp73. (**D-F**) IB shows the effects of *CBFB* (**D**), *RUNX1* (**E**), or *TP53* (**F**) deletion on ΔNp73 protein levels in MCF10A cells. (**G-H**) IB shows the restoration of TAp73 levels upon *CBFB* (**G**) overexpression (OE) in CBFB_KO MCF10A cells or *RUNX1* (**H**) inducible expression in RUNX1_KO MCF10A cells.

CBFB_KO and RUNX1_KO MCF10A cells, we re-expressed *TAp73* in CBFB_KO or RUNX1_KO MCF10A cells and then transplanted these cells into NSG mice. CBFB_KO and RUNX1_KO MCF10A cells transduced with an empty lentiviral vector formed tumors within 2 to 3 months, while CBFB_KO and RUNX1_KO MCF10A cells overexpressing TAp73 did not form tumors in any mouse up to 180 days (Table 1). This result demonstrates that the re-expression of TAp73 inhibits the tumorigenic ability of CBFB_KO and RUNX1_KO cells and further supports the notion that TAp73 is involved in the tumor suppressive function of CBFB.

## TAp73 loss and NOTCH3 overexpression cooperatively promote breast tumorigenesis

Previously, we have shown that the repression of *NOTCH3* is one of the mechanisms of the tumor suppressive functions of CBFB and RUNX1 [11]. After establishing TAp73 induction as another mechanism, we explored the relationship between TAp73 and the *NOTCH3* intracellular domain (NICD)—the active part of NOTCH3 involved in transcriptional regulation and oncogenic transformation of breast cancer. We first determined whether TAp73 and the NICD regulate the levels of each other. TAp73 overexpression in CBFB_KO and RUNX1_KO cells did not alter the protein levels of NOTCH3-NICD (Fig 5A and 5B). Conversely, TAp73 levels did not change in NOTCH3 deleted or overexpressed MCF10A cells and vice versa (Fig 5C–5E). These results demonstrate that TAp73 and NOTCH3-NICD do not regulate each other's expression.

Although NOTCH3-NICD OE alone is sufficient to transform MCF10A cells, its transformation capacity is weaker than CBFB or RUNX1 deletion [11]. Thus, we examined whether p73 loss cooperates with NOTCH3-NICD OE in promoting tumorigenesis. WT and p73_KO MCF10A cells were transduced with lentiviruses expressing the NICD of NOTCH3 (Fig 5E). The anchorage-independent assays showed that p73 loss enhanced the transformation ability of the NICD of NOTCH3, judged by both the percentage of transformed cells and the size of the colonies (Fig 5F and 5G). We next tested the cooperation between p73 depletion and NOTCH3-NICD OE in tumorigenesis *in vivo*. MCF10A cells carrying NICD OE alone, NICD OE + p73_KO, or p73_KO alone were transplanted into NSG mice. There was no tumor formation of MCF10A cells with p73_KO alone, consistent with the anchorage-independent assay. MCF10A cells with NICD OE or NICD OE + p73_KO grew tumors, and the tumor weight and size of the NICD OE + p73_KO group were significantly larger than those of the NICD OE group (Fig 5H and 5J). Therefore, p73 loss cooperates with NOTCH3 OE in breast tumorigenesis.

To gain insights into the role of p73 loss in tumor progression, we performed hematoxylin and eosin (H&E), Ki-67, and cleaved caspase 3 staining. H&E staining did not reveal an obvious difference in histology between NICD OE and NICD OE + p73 KO groups (S2A Fig). NICD OE + p73 KO tumors had a significantly higher Ki-67 signal than NICD OE tumors (Fig 5K and 5L), suggesting that the former group contained more proliferating cells. This result is also consistent with the observation that NICD OE + p73 KO group has a larger tumor size and weight than the NICD OE group (Fig 5H–5J). There was no statistically

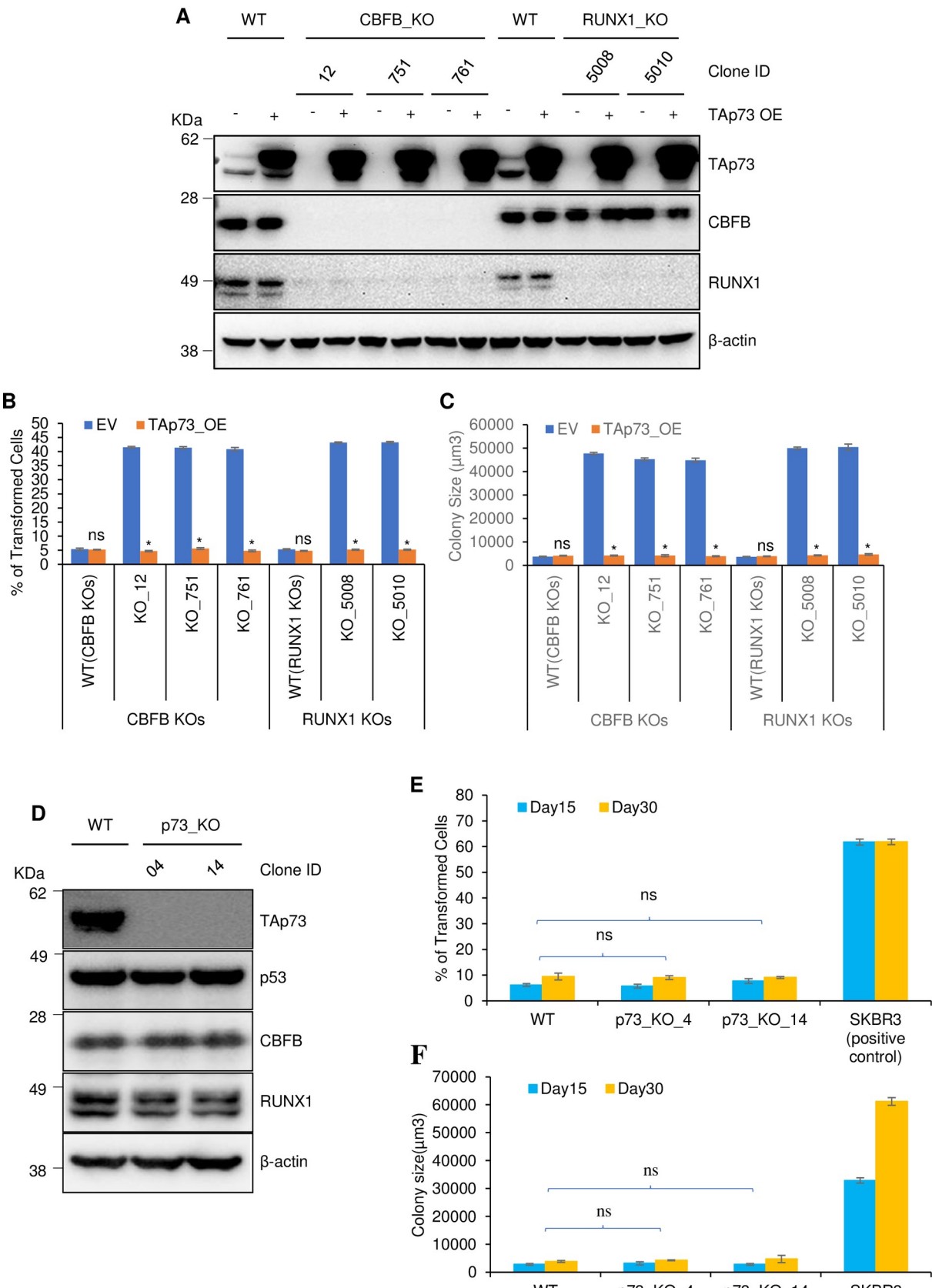

**Fig 4. TAp73 facilitates but is insufficient for cell transformation.** (**A**) IB shows the overexpression of TAp73 in CBFB_KO and RUNX1_KO MCF10A cells. Anchorage-independent growth assays show the effect of exogenously expressed TAp73 (TAp73_OE) on the percentage of transformed cells (**B**) and size (**C**) in CBFB_KO and RUNX1_KO MCF10A cells 15 days after plating. (**D**) IB shows the expression of TAp73, p53, CBFB, RUNX1, and β-actin in WT and p73_KO MCF10A cells. (**E-F**) Anchorage-independent assays show the percentage of transformed cells (**E**) and size (**F**) of WT, p73_KO MCF10A, and SKBR3 (positive control) cells. For **B**, **C**, **E**, **F**, Error bars are SEM; n = 3 biological repeats; p-values are from t-test (two-tailed, two-sample with equal variance); ns, not significant.

significant difference in cleaved caspase 3 signal between NICD OE and NICD OE + p73 KO groups (S2B and S2C Fig). Therefore, these results suggest that p73 loss promotes the proliferation of NICD-driven tumor cells.

## TAp73 loss is a common event in breast cancer cells

A previous report showed that *ΔNp73* is overexpressed in human breast tumors [41]. However, it remains unclear whether TAp73 is also dysregulated in breast cancer cells and tumors. To address this, we first surveyed the expression of TAp73 in two non-tumorigenic breast cell lines, MCF10A and MCF12A, and a panel of breast cancer cell lines. Although TAp73 was readily detected in MCF10A and MCF12A cells, it was undetectable in all the breast cancer cell lines tested (Fig 6A). These results show that TAp73 loss is common in cultured breast cancer cell lines.

To test whether TAp73 loss is a frequent event in human breast tumors, we examined the expression of TAp73 in the TCGA dataset (the METABRIC has no isoform information) (Fig 6B). Tumors in the dataset were divided into four groups: *TP53*_MUT;*CBFB*_MUT, *TP53*_MUT;*CBFB*_WT, *TP53*_WT;*CBFB*_MUT, and *TP53*_WT;*CBFB*_WT. TAp73 levels were significantly lower (p = 7.8e-08) in the *TP53*_MUT;*CBFB*_WT group (median = 19.93) than the *TP53*_WT;*CBFB*_WT group (median = 57.59). *TAp73* levels in the *TP53*_MUT;*CBFB*_MUT group (median = 43.85) and the *TP53*_WT;*CBFB*_MUT group (median = 28.24) were lower than that in the *TP53*_WT;*CBFB*_WT group (median = 57.57). However, the differences were not statistically significant, which may be due to the small number of tumors (3 for *TP53*_MUT;*CBFB*_MUT and 20 for *TP53*_WT;*CBFB*_MUT). Or the heterogeneous cell types (e.g., infiltrated stromal or immune cells) within tumors "contaminated" the *TAp73* read counts in bulk RNA-seq. We then performed immunohistochemistry (IHC) using a human tissue microarray containing normal breast tissues and breast tumors. We found that most breast tumors had much lower expression of TAp73, CBFB, and RUNX1 compared to adjacent normal breast tissues (Figs 6C, 6D, S3A and S3B). We did not find any significant correlation of TAp73 loss with any breast tumor subtypes (compare Figs 6D and S3C). In summary, our results suggest that TAp73 loss is a general feature of human breast cancer cell lines and tumors.

## Discussion

CBFB is an emerging tumor suppressor in human breast cancer [11,12]. The main goal of this study is to further elucidate the mechanisms of the tumor suppressive functions of CBFB in

**Table 1. Effect of TAp73 overexpression on the tumor formation of CBFB_KO and RUNX1_KO MCF10A cells.** Five million MCF10A cells of various genotypes, as indicated, were subcutaneously transplanted into NSG mice. Tumor incidence: the number of mice having tumor formation/total number of transplanted mice. Latency period: the time from tumor cell injection to tumor collection (endpoint defined by the animal study protocol).

| Genotype | Tumor incidence | Latency period (day) | Tumor size (diameter, mm) |
|---|---|---|---|
| CBFB_KO + Empty vector | 5/5 | 67±16.4 | 10.2±1.31 |
| CBFB_KO + TAp73 overexpression | 0/5 | No tumor formed 180 days after transplantation | ---- |
| RUNX1_KO + Empty vector | 5/5 | 73±13.4 | 9.8±3.25 |
| RUNX1_KO + TAp73 overexpression | 0/5 | No tumor formed 180 days after transplantation | ---- |

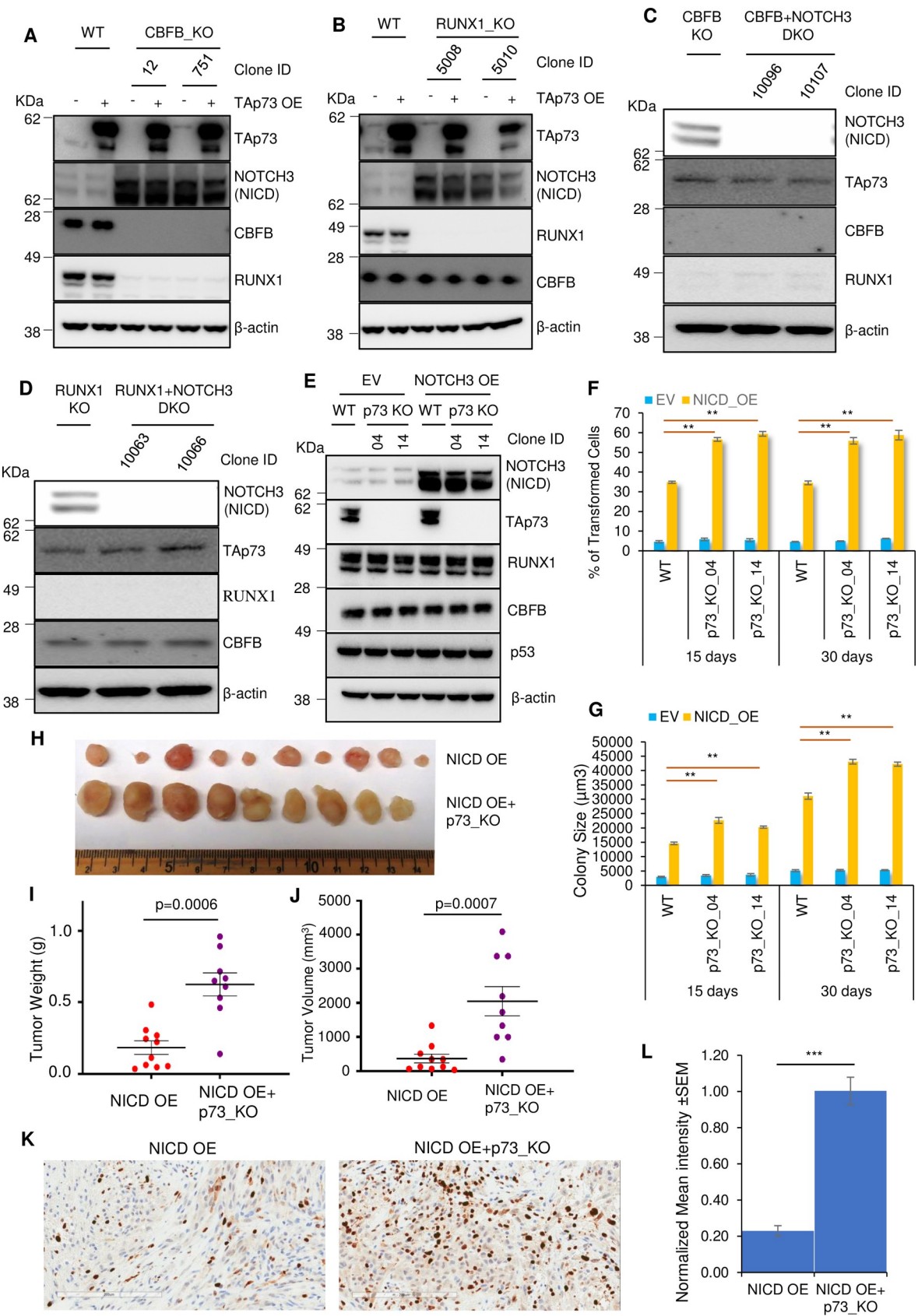

**Fig 5. TAp73 loss and NOTCH3 overexpression cooperatively promote breast tumorigenesis.** (**A-B**) IB shows the effect of TAp73 overexpression (OE) on NOTCH3 levels in CBFB_KO cells (**A**) or RUNX1_KO (**B**) MCF10A cells. (**C-D**) IB showing the effect of *NOTCH3* deletion on TAp73 in CBFB_KO (**C**) or RUNX1_KO (**D**) MCF10A cells. (**E**) IB shows the levels of NOTCH3(NICD), TAp73, RUNX1, CBFB, p53, and β-actin in WT, p73_KO clone 4, and clone 14 MCF10A cells with or without NOTCH3 (NICD) OE. (**F-G**) The anchorage-independent assay shows the cooperation of TAp73 loss and NICD OE in the transformation of MCF10A cells. **F**, % of transformed cells at day 15 and 30; **G**, size of colonies. EV, empty vector; NICD_OE, the NICD (NOTCH3) overexpression. Error bars = SEM; n = 3 biological repeats; **, p-value<0.01. p-values are from t-test (two-tailed, two-sample with equal variance). **H**, Xenograft tumors from MCF10A cells overexpressing the NICD of NOTCH3 or NICD OE with p73_KO grown in NSG mice for 70 days. **I-J**, Tumor weight (**I**) and volume (**J**) of tumors in **H**. p-values are from the Mann-Whitney test (two-tailed). **K**, Representative IHC images of Ki-67. Scale bar, 200 μm. **L**, Normalized mean intensity of Ki-67. See Materials and Methods for calculation of normalized mean intensity. Shown is the average of normalized mean intensity plus SEM from 15 images (5 randomly selected images from each of the three tumors). ***, p-value<0.001, t-test (two-tailed, two-sample with equal variance).

breast cancer. Our genetic analyses revealed the mutual exclusivity of *CBFB* and *TP53* mutations and suggested a functional association between CBFB and p53. To leverage the rich knowledge of the tumor suppressor p53, we investigated the molecular mechanism underlying this functional communication between CBFB and p53. Integrated genomic analyses identified *TAp73* as a common target of CBFB and p53 in normal human epithelial cells. We further demonstrated that TAp73 is one of the mediators for the tumor suppressive functions of CBFB and p53. Our data support a model described in Fig 7. The model states that CBFB/RUNX1 and p53 cooperatively activate the expression of *TAp73*. Mutation of either *CBFB* or *TP53* has a similar effect on the expression of *TAp73* as mutations of both genes. Thus, breast tumor cells gain little extra advantage by mutating both *CBFB* and *TP53* (Fig 7).

There are several caveats about our model. Several studies have recently shown that, in certain types of cancer or stem-like cells, TAp73 sustains tumor cell growth by regulating metabolism [29–31]. Therefore, it is likely that the roles of TAp73 in cancer depend on cancer types, specific populations within a given cancer type, and the unique genetic makeup of a tumor cell. Our results support the model that TAp73 is anti-tumorigenic in differentiated CBFB defective or NOTCH3-overexpressing breast cells. Another caveat is that the guide RNAs for p73 knockout delete both TAp73 and ΔNp73. Therefore, we cannot completely rule out the possibility that ΔNp73 also contributes to the anti-tumorigenic effect of p73 deletion. Moreover, *CBFB* mutations (about 5%) and *TP53* mutations (about 34%) cannot completely explain the widespread loss of TAp73 in breast cancer. It has been shown that dysregulation of other transcription factors, such as NRF2, and promoter methylation can lead to *TAp73* silencing in breast cancer[42].

Our previous study found that CBFB suppresses breast cancer partially through NOTCH3 repression [11]. In this study, we identified TAp73 as another mechanism for the tumor suppressive function of CBFB. Notably, TAp73 loss was insufficient for the transformation of breast cells. Instead, it facilitates the tumorigenic ability of NOTCH3 OE. Therefore, the tumor suppressive function of CBFB in breast cancer includes several mechanisms, *TAp73* activation, *NOTCH3* repression, and other yet unidentified genes or signaling pathways. In this regard, CBFB is very similar to p53, which regulates hundreds of targets involved in a plethora of cellular processes, such as apoptosis, cell cycle arrest, and DNA damage, to inhibit tumorigenesis [4]. However, deficiency in apoptosis, cell cycle arrest, and senescence alone does not fully recapitulate the tumor suppressive function of p53 [43]. Furthermore, besides TAp73, we identified 32 additional common targets of CBFB and p53, and many of them, such as *TGFB2*, are involved in tumorigenesis [44]. Therefore, some of these common targets also likely mediate the tumor suppressive functions of CBFB and p53.

Although our current model proposes that CBFB and p53 cooperatively induce the transcription of *TAp73*, it does not rule out the possibility that CBFB and p53 may crosstalk through other mechanisms. It is worth noting that CBFB is a dual-function protein—

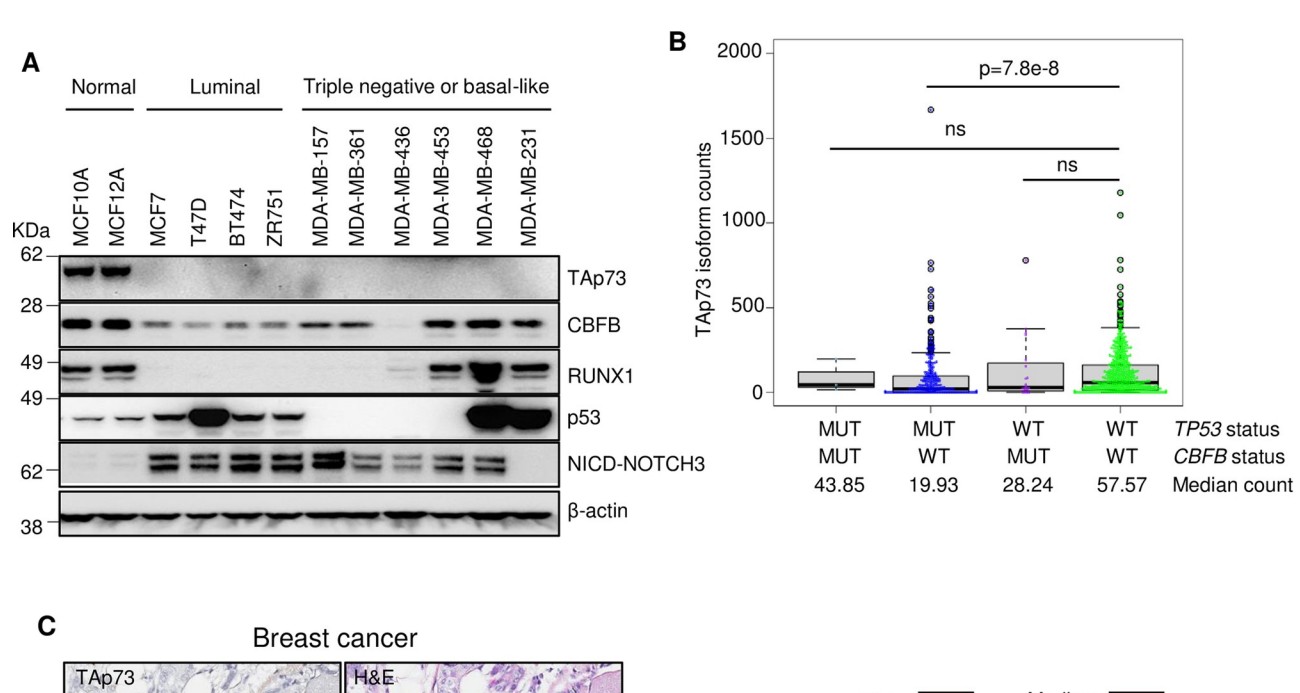

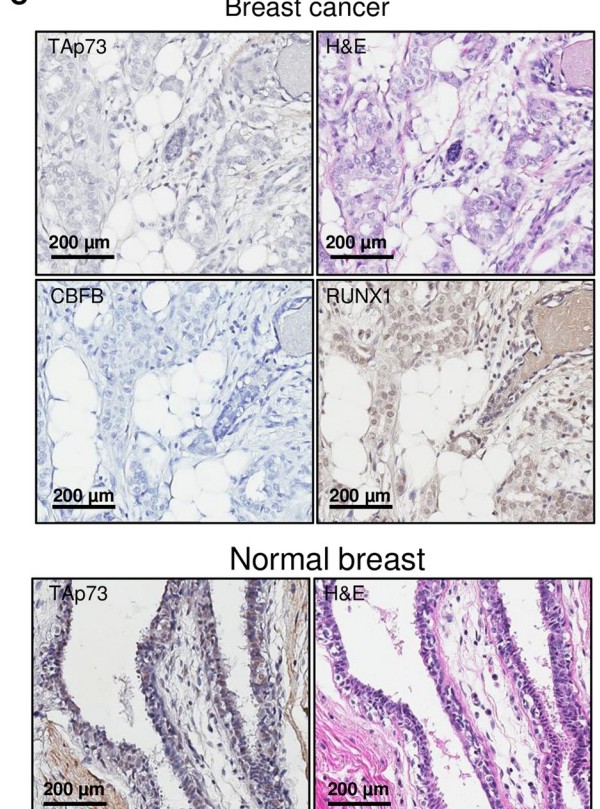

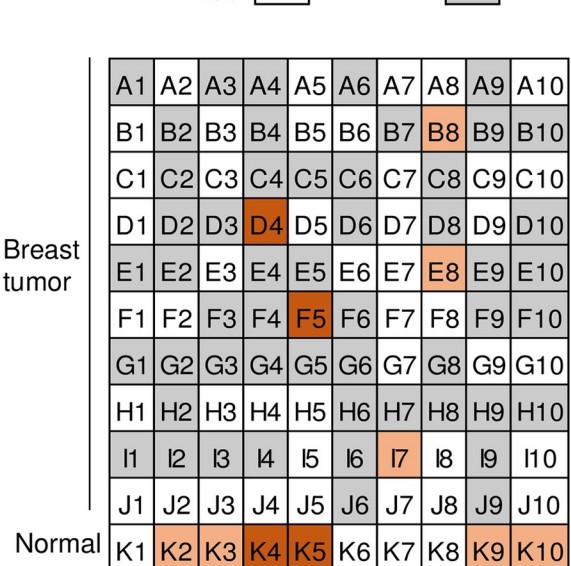

| TAp73 signal | Not detected | Low | Medium | High |
| :--- | :---: | :---: | :---: | :---: |
| **Sample type** | | | | |
| **Tumor** | 52/100 | 43/100 | 3/100 | 2/100 |
| **Normal** | 0/10 | 4/10 | 4/10 | 2/10 |

**Fig 6. TAp73 loss is a frequent event in human breast cancer cell lines and tumors.** (**A**) IB shows the expression of CBFB, RUNX1, p53, TAp73, and the NICD of NOTCH3 in multiple human breast cell lines. (**B**) Read counts of *TAp73* isoforms for each indicated genotype from bulk RNA-seq in the TCGA dataset. Median counts for each group were shown. ns, not significant. The p-value is from the Wilcoxon test. (**C**) Representative immunocytochemistry (IHC) images show the protein expression of TAp73, RUNX1, and CBFB in human normal breast and breast tumor tissue microarrays (TMAs). (**C**) Upper panel: Color codes show TAp73 IHC staining intensity in each tumor (core) of a human breast TMA. The total number of samples is 110, with 100 tumors and ten normal breast samples. Lower panel: a table summarizes the number of samples in each category: not detected, low intensity, medium intensity, and high intensity.

regulating translation in the cytoplasm and transcription in the nucleus [11]. The nuclear function of CBFB is mainly through its transcriptional partner, RUNX1. However, *RUNX1* and *TP53* mutations are not mutually exclusive, and the exact mechanism of this observation is unclear. One possibility is that other non-nuclear mechanisms underlying the mutual exclusivity of *CBFB* and *TP53* exist. Nonetheless, the finding that *TAp73* is a common target of CBFB and p53 significantly deepens our understanding of the tumor suppressive function of CBFB.

Our results may have important implications in the application of precision medicine for the treatment of breast cancer. As the NOTCH signaling pathway is frequently deregulated in breast cancer, targeting the NOTCH pathway has drawn much attention [45]. However, several clinical trials of testing the effect of inhibiting the NOTCH pathway in breast cancer had been either terminated or suspended [46]. One of the main reasons for these setbacks is the lack of a reliable biomarker to select patients who will likely benefit from NOTCH inhibition. Because NOTCH3 repression is a major response to CBFB loss-of-function, patients with breast tumors carrying *CBFB* mutations probably may benefit more from NOTCH inhibitors than those with tumors bearing wild-type *CBFB*. The cooperation between TAp73 loss and NOTCH3 OE raises the possibility of inhibiting breast tumorigenesis by simultaneously targeting TAp73 and NOTCH3. Targeting TAp73 may be achieved by identifying druggable genes or pathways regulated by TAp73 in the future. These interesting hypotheses are currently

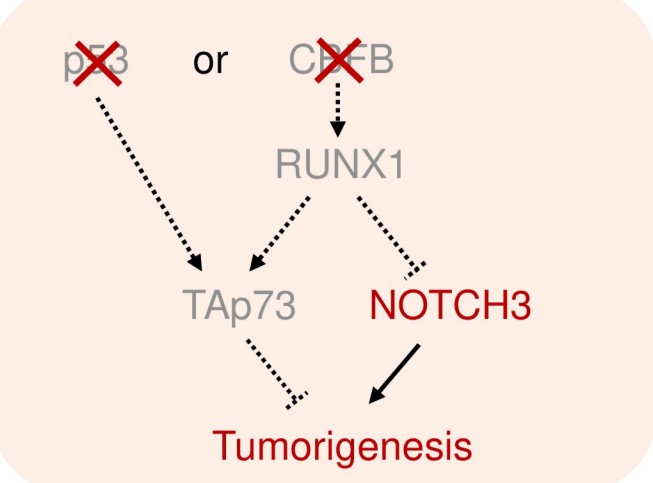

**Fig 7. A model shows the functional interaction between the CBFB and p53 in breast tumor suppression.** In normal breast cells, p53 and CBFB cooperatively maintain the expression of TAp73. In breast cancer cells, either CBFB loss or p53 loss reduces the levels of TAp73, which contributes to breast cell transformation and tumorigenesis by cooperating with NOTCH3 overexpression. TAp73 is one of the mechanisms underlying the functional interaction between CBFB and p53. Grey: loss or reduced activity; dashed lines: dysregulation; Arrows; induction; hammerheads: repression.

being investigated in our laboratory and have the potential of generating new therapeutic strategies in breast cancer precision medicine.

## Materials and methods

### Ethics statement

All animal studies were performed under the guidelines of the Institutional Animal Care and Use Committee (IACUC)-approved protocols at the National Cancer Institute (NCI) and National Heart, Lung, and Blood Institute (NHLBI).

### Genetic interaction analysis of gene mutations in human breast cancer

In the METABRIC and TCGA datasets, we calculated the genetic interactions between mutations in *CBFB* and other genes using Fisher's exact test and reported only those interactions with a p-value less than 0.1. Genetic interaction between two gene mutations with an odds ratio larger than one is considered co-occurring, while interaction with an odds ratio less than one indicates mutually exclusive. Oncoprints were generated using cBioPortal or R [36].

### Cell culture

MCF10A, MCF7, BT474, T47D, ZR751, MDA-MB-157, MDA-MB-231, MDA-MB-361, MDA-MB-436, MDA-MB-453, MDA-MB-468, and HEK-293T were purchased from ATCC (Manassas, VA). MCF12A cells were a kind gift from Dr. Stefan Ambs (NIH, Bethesda). Immortalized human mammary epithelial cells MCF10A and MCF12A cells were cultured in DMEM/F12 media supplemented with 5% horse serum, 10 μg/ml human recombinant insulin (Sigma, Cat# 910077C), 20 ng/ml human EGF (PeproTech, Cat# AF100-15), 500 ng/ml hydrocortisone (Sigma, Cat# H0888g), 100 ng/ml cholera toxin (Sigma, Cat# C8052-5MG) and antibiotics. ATCC formulated Eagle's Minimum Essential Medium supplemented with 10% FBS was used to culture MCF7 cells. All other cell lines were cultured in DMEM media supplemented with 10% FBS and antibiotics. All cell lines were maintained at 37˚C in an incubator supplied with continuous 5% $CO_2$.

### Antibodies

The antibodies used for immunoblotting were: p53 (DO-1, Santa Cruz, Cat# sc-126, at 1:1000 dilution), CBFB (Bethyl, Cat# A303-547A, at 1:1000 dilution), RUNX1 (Cell Signaling, Cat# 4334s, at 1:1000 dilution), NOTCH3 (Cell Signaling, Cat# 5276s, at 1:1000 dilution), TAp73 (Novus Biologicals, Cat# NBP2-24737, at 1:1000 dilution), ΔNp73 (Novus Biologicals, Cat#NBP2-24873, at 1:1000 dilution), and β-actin (Sigma, Cat# A5316, at 1:5000 dilution). The antibodies used for chromatin immunoprecipitation (ChIP) were: p53 (DO-1 Santa Cruz, Cat# sc-126, 10 μg) and RUNX1 (Abcam, Cat# ab23980, 10 μg).

### Immunoblotting

Cells were lysed in the whole-cell lysis buffer (25 mM Tris-HCL, pH 7.4, 150 mM NaCl, 1% NP-40, 1% Sodium deoxycholate) followed by 5-minute sonication (30 seconds on and 30 seconds off) at 4˚C. Protein concentration was determined by the Bradford assay, and the same amount of proteins were resolved on 4–12% Bis-Tris NuPAGE Protein gels (Life Technologies) and transferred to nitrocellulose membranes (Bio-Rad). After one hour of blocking in 5% non-fat milk in PBS-Tween 20, the membranes were incubated overnight at 4˚C with primary antibody. The next day, the membrane was washed three times before being incubated with a secondary antibody. All images were captured in a ChemiDoc MP Imaging System (Bio-Rad)

using the ImageLab Touch Software (Version 2.2.0.07). A distinct feature of the NuPAGE gel system is that most proteins' apparent molecular weights are different from the calculated or known molecular weight. The molecular weights shown in the figures are the SeeBlue Plus2 pre-stained protein standard (Life Technologies, Cat# LC5925) in the NuPAGE system per the manufacturer's instruction.

### ChIP-seq, RNA-seq, and data analysis

ChIP-seq and RNA-seq were performed as described previously [11,47]. DNA and RNA samples were sent to the Next-Generation Sequencing Facility at the Center for Cancer Research (CCR) at NCI for deep sequencing. For ChIP-seq, we used 10 ng precipitated DNA. The MACS algorithm was used to identify peaks [48]. For RNA-seq, we used 1 μg of total RNA, which was subsequently subject to rRNAs removal, size selection, cluster generation, and deep sequencing on the NextSeq 500 platform. Common transcriptional targets of CBFB and RUNX1 were identified in our previous study[11]. To identify p53 targets, we obtained four biological repeats for WT, p53_KO clone 313, and clone 414. The DESeq2 algorithm [49] was used to identify p53-regulated transcripts at cutoffs of FDR (false discovery rate) of 0.05 and fold change of 2. In the analysis, four repeats of clone 313 and four repeats of clone 414 were combined as eight repeats of p53_KO and compared to the four repeats of WT.

### Cloning RUNX1 response element in the *TP73* gene and the Reporter assay

A fragment of 501-bp DNA, termed full-length response element (FL_RE), was amplified using PCR and oligos: jh5975 and jh5978 (S1 Table). The PCR amplified FL_RE was digested with XhoI and Hind III and cloned into pGL4.23_Luc2_minP (Promega Corporation, Cat# E8411), which encodes the *firefly* luciferase gene from *Photinus pyralis*. Using FL_RE as the template, we generated truncated RE fragment1 (175 bp, jh5975/jh5976) and fragment2 (431 bp, jh5975/jh5977). We found three putative RUNX1 motifs within fragment2. Afterward, site-directed mutagenesis was utilized to mutate these three putative motifs, and the resulting fragments were named mut1, mut2, and mut3. In addition, combinations of these mutant response elements were made to pinpoint the motif(s) that is (are) responsible for the regulation of *TP73* by RUNX1. All sequences of oligos are in the S1 Table.

The *firefly* reporters (1 μg) containing the RUNX1 response element were co-transfected with a reporter (200 ng) expressing the *Renilla* luciferase gene (from *Renilla reniformis*) into MCF10A cells plated in a well of 6-well plates using lipofectamine 2000. After 24 hours, cells were lysed, and normalized luciferase activity (*firefly* versus *Renilla*) was measured as previously described [50–52]. Normalized luciferase activity using an empty pGL4.23_Luc2_minP and an empty expression vector was set as 100%.

### Lentiviral vector cloning and virus production

HA-p73α-pcDNA3 was a gift from William Kaelin (Addgene plasmid # 22102; http://n2t.net/addgene:22102; RRID: Addgene_22102). The p73 (TAp73) cDNA was cloned into the pENTR-TOPO-D vector (Life Technologies). The resulting pENTR-TOPO-TAp73 entry vector was recombined with pLenti6/V5-DEST vector (Life Technologies) to generate a lentiviral vector expressing *TAp73*. Lentiviruses were packaged in HEK-293T cells, as described previously [11,47]. Cells were incubated with lentivirus supernatant and 6 μg/ml polybrene overnight before replacing them with fresh growth media.

## CRISPR knockout cell lines

MCF10A cell lines with *CBFB*, *RUNX1*, and *NOTCH3* deletion (knockout) were generated and described previously [11]. MCF10A cell lines with *TP53* deletion (clone 313 and 414) were generated in this study using the CRISPR-Cas9 system and guide RNAs as described previously [53]. Briefly, to generate clone 313, we transfected MCF10A cells with two plasmids expressing two guide RNAs (corresponding DNA sequences: 5'-GGGCAGCTACGGTTTCC GTC-3' and 5'-GCATCAAATCATCCATTGCT-3') together with an EGFP-expressing vector. Flow cytometry was used to sort EGFP-positive cells, which were then plated at a single-cell density. Single clones were picked, propagated, and examined for p53 loss using Western blotting. To generate p53_KO clone 414, we used another pair of plasmids expressing two different guide RNAs (corresponding DNA sequences: 5'-GGATGATTTGATGCTGTCCC-3' and 5'-GACGGAAACCGTAGCTG-CCC-3'). All these guide RNAs for p53 deletion were cloned into the vector pX330 (a gift from Feng Zhang (Addgene plasmid # 42230; http://n2t. net/addgene:42230; RRID:Addgene 42230) [54]. To delete p73, we generated two gRNAs (5'-GCACCTTCGACACCATGTCGC-3' and 5'-GAGGCCGCGCGGCTGCTCATC-3') and cloned them into the vector lentiCRISPRv2 (a gift from Brett Stringer (Addgene plasmid# 98290; http://n2t.net/addgene:98290; RRID:Addgene 98290).

## Anchorage-independent growth assay

We prepared a lower layer of 0.5% agarose (Sea Kem LE Agarose, Cat# 50004) in a culturing medium in a 6-well plate. After the lower layer was completely solidified, we mixed 2,500 single MCF10A cells with warm (37˚C) 0.35% agarose in a culturing medium and quickly added the mixture on top of the 0.5% agarose lower layer. Colonies were counted and imaged with a 20X magnification on an Axiovert 25 microscope (Zeiss)15 and 30 days after plating. Transformation percentage was calculated by dividing the number of colonies formed by the total number of plated cells. The colony size was calculated based on images of twenty randomly selected colonies from duplicate wells. Three biological repeats were performed for each anchorage-independent growth assay.

## Animal studies

For xenograft assays: 5 million MCF10A cells were resuspended in 100 μl DMEM/F12 medium supplemented with 25 mM HEPES (pH 7.4). 50 μl Matrigel was added before these cells were subcutaneously injected into 8-week old female NSG (NOD-scid-gamma) mice (The Jackson Laboratory, strain: 005557). Tumors were harvested based on the criteria pre-determined in the animal study protocols approved by the IACUC committees— when tumors reach 2 cm in diameter or the host experiences severe health issues, such as difficulty breathing or movement or loss of more than 15% body weight. Tumors were weighed, measured, cut into small pieces, and fixed in 10% neutral-buffered formalin for 16 hours. Subsequently, tumors were stained by hematoxylin and eosin at Histoserv, Inc. (Germantown, MD, USA).

## Human breast cancer and normal breast tissue microarray

Human tissue microarrays (TMA) of breast cancer and normal breast tissue were purchased from US Biomax, Inc. (Cat# BC081120c). This breast cancer TMA contains 110 cases/110 cores: 100 cases of invasive ductal carcinoma and 10 adjacent normal breast tissue.

## Immunohistochemistry

For TMA, glass slides having formalin-fixed paraffin-embedded tissue sections were first deparaffinized using xylene and then sequentially hydrated using decreasing percentages of alcohol from 100% to 50%. After hydration, antigens were retrieved by boiling in 10 mM sodium acetate for 25 minutes. Subsequently, after cooling, incubation with 3% $H_2O_2$ for 10 minutes was used to inactivate endogenous peroxidases. Further, slides were washed with PBS containing 0.1% Tween-20 and blocked with goat serum and Fc-blocker. After blocking, samples were incubated with primary antibodies overnight [CBFB (Bethyl, Cat# A393-549A, at 1:100 dilution), RUNX1 (Abcam, Cat# ab23980, at 1:100 dilution), and p73 (Abcam, Cat# ab40658, at 1:100 dilution)]. The next day, slides were extensively washed with PBS and incubated with a biotinylated goat anti-rabbit secondary antibody from the VECTASTATIN ABC HRP kit (Vector Laboratories, Cat# PK-4001) as per the manufacturer's instructions. Afterward, slides were incubated in biotin-avidin solution, and the color was developed using the DAB system (Vector Laboratories, Cat# SK-4100). Slides were dehydrated, cleared, and mounted using VectaMount Mounting Medium (Vector Laboratories, Cat# H-5000).

For Ki-67 and cleaved caspase-3 staining, FFPE sections were stained on the Bond RX autostainer (LeicaBiosystems). Briefly, following antigen retrieval with Bond ER1, sections were incubated for 30 minutes with Ki-67 (Cell Signaling Technology, Cat#: 12202, diluted 1:200) or cleaved caspase-3 (Cell Signaling Technology, Cat#: sc-9661, diluted 1:100) antibody. Staining was completed with Bond Polymer Refine DAB Detection Kit, with the omission of the Post Primary Reagent.

Scanned images were exported from the Aperio ImageScope and imported into Fiji [55] for quantification. Five randomly selected 20X images for each tumor (total 3) were color deconvoluted using the H DAB setting. Background staining was removed by setting channel-specific thresholds. Mean intensities of nuclear staining and antibody staining were measured, and the normalized mean intensities (antibody staining versus nuclear staining) were calculated.

## Statistical analysis

Mutual exclusivity of mutations in CBFB and other genes was analyzed using Fisher's exact test. Tumor mutation load in breast cancer patient samples was analyzed using the Wilcoxon test. Quantification of the intensity of protein bands in western blots was performed using ImageJ software (NIH). Soft agar colony formation assay and p73 response element luciferase assay were analyzed using the two-tailed t-test, assuming equal variance in the two groups. Tumor weight and tumor volume in xenograft assays were analyzed using the non-parametric Mann-Whitney test. Expression of *TAp73* isoforms in human breast cancer patient samples bearing *TP53* or/and *CBFB* mutations was compared with that in the wild type (for both *TP53* and *CBFB*) group using the Wilcoxon test. IHC images of p73 in human TMA were scored for high, medium, low, or not detected staining based on staining intensity and fraction of positively stained tissue. For normal breast tissue, the adipose tissue area was excluded for analysis. Representative images for different scoring were provided in S3B Fig.

## Supporting information

**S1 Fig. Re-expression of p73 reverses the transformation of CBFB and RUNX1 KO cells.** (**A & B**) Representative images of anchorage-independent growth assays to show the effect of exogenously expressed TAp73 (p73_OE) on colony formation in CBFB_KO (**A**) and RUNX1_KO cells (**B**). **C,** Percentage of transformed cells, as evaluated by the number of colonies formed, in CBFB_KO and RUNX1_KO cells after overexpression of TAp73, 30 days after

plating. Error bars are SEM, n = 3 (biological repeats); one asterisks, p-value <0.01, ns, p-value >0.05 (empty vector, EV versus TAp73 OE). The t-test is two-tailed, two-sample equal variance. **D,** Size of colonies in CBFB_KO and RUNX1_KO cells after overexpression of TAp73, 30 days after plating. Error bars are in SEM, n = 3 (biological repeats); one asterisks, p-value <0.01; ns, p-value >0.05 (EV versus TAp73 OE). The t-test is two-tailed, two-sample equal variance.
(TIF)

**S2 Fig. H&E and cleaved caspase 3 staining of xenografts of NICD OE and NICD OE+p73 KO.** (**A**) H&E staining. Representative images from two tumors were shown. Scale bar, 100 μm. (**B**) IHC of cleaved caspase 3. Scale bar, 200 μm. (**C**) Normalized mean intensity of cleaved caspase 3. See Materials and Methods for calculation of normalized mean intensity. Shown is the average of normalized mean intensity ± SEM from 15 images (5 randomly selected images from each of the three tumors). P-value is from t-test (two-tailed, two-sample equal variance).
(TIF)

**S3 Fig. TAp73 is downregulated in human breast tumors.** (**A**) TAp73 IHC images in a human breast tumor tissue microarray (TMA), which includes 10 normal breast tissue samples at the bottom. (**B**) Scoring examples of TAp73 IHC staining for high, medium, low, and not detected in TMA. (**C**) Subtypes of breast tumor samples in the TMA: Estrogen receptor (ER)/ Progesterone receptor (PR) positive, ER/PR, and HER2 receptor-positive, HER2 receptor-positive and triple-negative.
(TIF)

**S1 Table. Oligo sequences.**
(XLSX)

## Acknowledgments

We want to thank Bao Tran's Next-Generation Sequencing Facility at the Center for Cancer Research (CCR) for RNA-seq and ChIP-seq, and the Office of Science and Technology Resources (OSTR) at CCR, NCI. We thank Brandi Carofino for providing scientific editing assistance during the preparation of this manuscript.

## Author Contributions

**Conceptualization:** Jing Huang.

**Data curation:** Navdeep Malik, Hualong Yan, Wendy DuBois, Yu-Chou Tseng, Young-Im Kim, Shunlin Jiang, Chengyu Liu, Maxwell Lee, Jing Huang.

**Formal analysis:** Navdeep Malik, Hualong Yan, Howard H. Yang, Gamze Ayaz, Wendy DuBois, Maxwell Lee, Jing Huang.

**Funding acquisition:** Jing Huang.

**Investigation:** Navdeep Malik, Hualong Yan, Jing Huang.

**Methodology:** Navdeep Malik, Maxwell Lee.

**Supervision:** Jing Huang.

**Validation:** Jing Huang.

**Writing – original draft:** Navdeep Malik, Jing Huang.

**Writing – review & editing:** Jing Huang.

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
