## [Decision Letter · Decision Letter 0]

8 Feb 2021

Dear Dr Huang,

Thank you very much for submitting your Research Article entitled 'CBFB cooperates with p53 to maintain TAp73 expression and suppress breast cancer' to PLOS Genetics.

The manuscript was fully evaluated at the editorial level and by independent peer reviewers. The reviewers appreciated the attention to an important problem, but raised some substantial concerns about the current manuscript. Based on the reviews, we will not be able to accept this version of the manuscript, but we would be willing to review a much-revised version. We cannot, of course, promise publication at that time.

If you decide to revise the manuscript for further consideration at PLOS Genetics, please aim to resubmit within the next 60 days, unless it will take extra time to address the concerns of the reviewers, in which case we would appreciate an expected resubmission date by email to plosgenetics@plos.org.

[LINK]

We are sorry that we cannot be more positive about your manuscript at this stage. Please do not hesitate to contact us if you have any concerns or questions.

Yours sincerely,

Peng Jiang

Guest Editor

PLOS Genetics

Peter McKinnon

Section Editor: Cancer Genetics

PLOS Genetics

**Comments to the Authors:**

Reviewer #1: This manuscript by Malik N. et al reports that CBFB cooperates with p53 to maintain TAp73 expression and suppress breast cancer. Authors find that CBFB and TP53 mutations are mutually exclusive in breast cancer, and CBFB regulates p73 expression to display its tumor suppressive function in breast cancer. These results provide a new mechanism of CBFB in suppression of breast cancer. The findings from this manuscript are interesting and original. Overall experiments are well designed and results support the major conclusions.

Some minor points:

1. Fig 1F: it appears that p53 KO slightly reduces the CBFB levels. Authors may want to confirm this result or modify their statement that p53 does not affect CBFB expression in breast cell lines.

2. Authors found that p53 and CBFB can regulate a set of genes in the same direction. p73 was proved to be an important downstream gene that mediates CBFB function in tumor suppression in breast cancer. Authors may want to discuss whether it is possible that some other genes in their list also contribute to the function of CBFB in breast cancer.

3. As authors described, p73 expression is frequently lost in breast cancer. It will be nice if authors can discuss other mechanisms contributing to p73 loss in breast cancer in addition to CBFB mutations given that CBFB mutations are not very common in breast cancer (around 5%).

Reviewer #2: The manuscript by Malik et al. describes that CBFB and p53 co-regulates TAp73 expression to suppress breast cancer. In addition, they showed that TAp73 re-expression reduces the tumorigenicity of CBFB-depleted cells and that TAp73 loss enhanced the tumorigenic effect of NOTCH3 overexpression. The co-regulation of TAp73 by CBFB and p53 would be important for understanding not only p53-mediated gene regulation but also tumor-suppressive function of CBFB. In addition, it provides a new mechanistic insight into how the tumor suppressor gene TAp73 is regulated. Therefore, these findings are obviously of interest. However, the below comments need to be addressed to improve this manuscript before its publication.

1) As authors may be aware, mutual exclusivity of two gene mutations in cancer may result from cancer cell death that is caused by co-occurrence of two gene mutations. Several examples are described in following articles: 1) Cisowski and Bergo, 2017 [PMID: 27416373]; 2) Zhao et al 2017 Nature [PMID: 28166537]). In this regard, although it is plausible that mutual exclusivity of CBFB and TP53 mutations would implicate a functional link between CBFB and TP53, it would not be reasonable to mention that the co-activation of the tumor suppressor gene TAp73 by CBFB and p53 would provide a mechanistic explanation to the mutual exclusivity of CBFB and TP53 mutations. This is because TAp73 downregulation that is caused by CBFB or TP53 depletion did not result in cancer cell death but increased tumorigenicity of NOTCH3-overexpressed MCF10A cells.

Therefore, they need to reword (or delete) multiple sentences throughout the text. Examples include but not are limited to the following ones: Abstract (therefore providing a mechanistic explanation to the mutual exclusivity of CBFB and TP53 mutations, page 2); discussion (This model explains the mutual exclusivity of CBFB and TP53 mutations, page 15); and Fig 7 legend (This model can explain the mutual exclusivity of CBFB and TP53 mutations in breast tumors).. In addition, the bottom panel in Fig. 2A may not be needed.

2) In abstract, authors mention “a functional interaction between CBFB and p53”. In addition, authors mention “suggesting that CBFB functionally interacts with p53 in breast cancer cells” in page 6. There are similar statements. “Functional interaction” and “functionally interacts” might be too specific. Authors should use a different word.

3) In abstract, authors introduce NOTCH3 without an explanation. It would be helpful for readers if they mention like “overexpression of NOTCH3, a CBFB-suppressed oncogene” instead of “NOTCH3 overexpression”

4) In Fig. 2D, where are the p53’s consensus binding sequences? Please mark them.

5) ChIP-seq data for p53 and RUNX1 are good results. To examine whether p53 and RUNX1/CBFB cooperate to regulate TAp73, authors need to show a more convincing result. Does p53 activate the FL-RE reporter (relevant to Fig. 2E)? Does the co-transfection of p53 and RUNX1 (or CBFB) activate reporter gene expression more than single transfection?

6) Authors need to show data (that may be similar to the bottom panels in Fig. 1A & 1B) for epistasis analysis in ER-positive breast tumors.

7) In Fig 3B-3H, there are numbers (e.g., 12, 751 and 761) as labels. Although these numbers look clone numbers, authors should clearly indicate what these are in legends or figures.

8) It seems unusual to use TS for tumor suppressive.

9) In page 11, the reduction TAp73 expression upon CBFB or RUNX1 deletion would better be “the reduced TAp73 expression….” or “the reduction in TAp73 expression….”

Reviewer #3: In this manuscript, the authors unmask a novel tumour suppressor mechanism in breast cancer, which is built around the CBFB-TAp73 axis. The authors find that CBFB and TP53 mutations are mutually exclusive in breast cancer. Further analysis suggest that p53 and CBFB converge towards a common regulation of the TAp73 gene. Indeed, CBFB maintain TAp73 expression in the MCF10A breast cells. TAp73 regulation has important functional implications. Indeed, loss of TAp73 cooperates with NOTCH3 overexpression in augment if breast cancer cell tumorigenicity in vitro and in vivo. A tumour suppressive role for TAp73 is confirmed by its loss of expression in a subset of breast cancers.

I found this paper to be well-written and quite interesting to read. Some of the findings are intriguing and of potential relevance for the scientific community. I have few observations, which I hope the authors will find helpful.

Initially the authors exclude any direct regulation between p53 and CBFB. They support this conclusion with the western blot presented in Figure E and F. I assume it will not come as a surprise to the authors if I do not fully agree with their interpretation. Indeed, it seems that CBFB levels are increased in p53 KO clones. I would appreciate if the authors could reassess this experiment. Perhaps a quantification of the western blot bands (using a less exposed actin) could be informative. To be honest, even if my observation of the data should turn out to be correct, I do not feel that this would detract from the strengths of the manuscript and I would still endorse its publication. It might however entail some degree of rewriting. I was also wondering why the authors observe two bands for CBFB protein in panel F and a single band in panel E.

I shall confess that this reviewer is not an expert of ChIP-seq. However, at least from the data presented in figure 3A, it seems that the H3K4me3 signals is decreased in the DNp73 promoter in p53 KO cells, which would contradict the authors’ interpretation of the results. I would appreciate if the authors could comment on this. Have the authors picked any signal from the H3K4me3 in the intragenic region that harbours the RUNX1 binding sites?

The luciferase experiments are quite convincing and show a thorough analysis of the RUNX1 bindings sites, although I would argue that site 1 seems to be quite irrelevant compared to the impact of mutating the site 2 and 3. Perhaps statistical analysis of the Luc experiments could help discerning the impact of the different site mutation, especially site 1. In addition, I was wondering whether the authors could ascertain if the response of the luciferase construct to RUNX1 is abolished in CBFB KO cells. I am asking this on the light of the data reported in Figure 3H, where RUNX1 overexpression restores TAp73 levels in CBFB KO cells, indicating that RUNX1 might regulate p73 expression at least partially independently of CBFB.

The xenograft experiments are interesting. I was hoping that the authors could provide an H&E assessment of those tumours. It would be interesting to see if loss of p73 alters tumours histology. Similarly, it would be nice, though not essential, if the authors could assess expression of ki67 and cleaved caspase 3 by IHC in those tumours as an indication as to whether p73 loss affect proliferation/survival.

With regard to loss of TAp73 in human breast cancer, I am well aware of the suboptimal performances of most antibodies against p73 proteins. However, it would important to provide clear example images of the scoring presented in Figure 6D (high, medium, low…). Finally, would it be possible for the authors to retrieve information about TAp73 expression in the breast cancer datasets and compare expression levels in CBFB/p53 wild-type vs CBFB mutated tumours? I appreciate this might not be possible, but it could provide a nice confirmation of the interaction between CBFB and TAp73 in human samples.

The p73KO cells seem to be knockout for the whole gene and not selective for TAp73. Perhaps the authors could elaborate on this understandable limitation in the discussion section.

The author should expand the materials and methods section providing a paragraph detailing the statistical analysis approached used in the manuscript. Similarly, an accurate description of the methodology used for the TMA scoring is necessary.

MINOR POINTS

I would appreciate that, if this is in agreement with the journal guidelines, the authors could explain figure legends to include details of descriptive and inferential statistical analysis for the different graph/data reported.

I would also be grateful if the authors could indicate the bench mark molecular weight in the western blots.

Honestly, I do not see the point of Figure 1D and Figure 2A. The author’s hypotheses, in both cases, are clearly outlined in the main text.

The description of the generation of p73KO cells seems to have been written twice in the methods section, with a minor change in one of the gRNA sequence. Could the author please check this?

One aspect that I like about this manuscript is that it describes TAp73 in its “original role” as tumour suppressor gene. Indeed, this is in contrast with recent reports indicating that TAp73 could paradoxically sustain tumour progression, especially through regulation of tumour cellular metabolism. Notably, this has also been reported for breast cancer stem cells. It would be interesting if the authors could comment their results on the light of these works. Indeed, their observations would suggest that the impact of TAp73 expression could depend on the subtype of breast cancer, with CBFB/NOTCH3-driven cancer highlighting a TS function for the TAp73 gene. However, although intrigued, I would understand if the authors should feel that this discussion might not be relevant to their analysis.

**Have all data underlying the figures and results presented in the manuscript been provided?**

Reviewer #1: Yes

Reviewer #2: Yes

Reviewer #3: Yes

PLOS authors have the option to publish the peer review history of their article (what does this mean?). If published, this will include your full peer review and any attached files.

Reviewer #1: No

Reviewer #2: No

Reviewer #3: No

---

## [Decision Letter · Decision Letter 1]

14 Apr 2021

Dear Dr Huang,

We are pleased to inform you that your manuscript entitled "CBFB cooperates with p53 to maintain TAp73 expression and suppress breast cancer" has been editorially accepted for publication in PLOS Genetics. Congratulations!

Yours sincerely,

Peng Jiang

Guest Editor

PLOS Genetics

Peter McKinnon

Section Editor: Cancer Genetics

PLOS Genetics

Comments from the reviewers (if applicable):

Reviewer's Responses to Questions

**Comments to the Authors:**

Reviewer #1: Authors have addressed my comments. This revised manuscript has been substantially improved. I will suggest the acceptance of this revised manuscript.

Reviewer #2: Authors has satisfactorily addressed all the questions of this reviewer. I would like to recommend this manuscript for its publication in PLOS Genetics.

Reviewer #3: I would like to thank the authors for their consideration and efforts to answer my comments. I am really pleased with the ki67 data and sorry that the dataset analysis has been unfortunately unsuccessful. But again, I really appreciated the effort and I can only confirm that this is a nice manuscript that deserves publication.

**Have all data underlying the figures and results presented in the manuscript been provided?**

Reviewer #1: Yes

Reviewer #2: Yes

Reviewer #3: None

PLOS authors have the option to publish the peer review history of their article (what does this mean?). If published, this will include your full peer review and any attached files.

Reviewer #1: No

Reviewer #2: No

Reviewer #3: No

**Data Deposition**

http://datadryad.org/submit?journalID=pgenetics&manu=PGENETICS-D-21-00010R1

**Press Queries**

---

## [Editor Report · Acceptance letter]

30 Apr 2021

PGENETICS-D-21-00010R1 

CBFB cooperates with p53 to maintain TAp73 expression and suppress breast cancer 

Dear Dr Huang, 

We are pleased to inform you that your manuscript entitled "CBFB cooperates with p53 to maintain TAp73 expression and suppress breast cancer" has been formally accepted for publication in PLOS Genetics! Your manuscript is now with our production department and you will be notified of the publication date in due course.

With kind regards,

Katalin Szabo

PLOS Genetics

On behalf of:
